# C1q and Tumor Necrosis Factor Related Protein 9 Protects from Diabetic Cardiomyopathy by Alleviating Cardiac Insulin Resistance and Inflammation

**DOI:** 10.3390/cells12030443

**Published:** 2023-01-29

**Authors:** Ricarda Haustein, Felix A. Trogisch, Merve Keles, Susanne Hille, Manuela Fuhrmann, Nina Weinzierl, Shruthi Hemanna, James Thackeray, Yanliang Dou, Carolin Zwadlo, Natali Froese, Julio Cordero, Frank Bengel, Oliver J. Müller, Johann Bauersachs, Gergana Dobreva, Joerg Heineke

**Affiliations:** 1Department of Cardiology and Angiology, Hannover Medical School, 30625 Hannover, Germany; 2Department of Cardiovascular Physiology, European Center for Angioscience (ECAS), Medical Faculty Mannheim, Heidelberg University, 68167 Mannheim, Germany; 3Department of Internal Medicine III, University Hospital Schleswig-Holstein, 24105 Kiel, Germany; 4Department of Nuclear Medicine, Hannover Medical School, 30625 Hannover, Germany; 5Cardiovascular Genomics and Epigenomics, ECAS, Medical Faculty Mannheim, Heidelberg University, 68167 Mannheim, Germany; 6German Center for Cardiovascular Research (DZHK), Partner Site Partner Site Hamburg/Kiel/Lübeck, 24105 Kiel, Germany; 7DZHK, Partner Site Heidelberg/Mannheim, 69120 Heidelberg, Germany

**Keywords:** high-fat diet, cardiac endothelial cells, diastolic dysfunction, paracrine signaling

## Abstract

(1) Background: Diabetic cardiomyopathy is a major health problem worldwide. CTRP9, a secreted glycoprotein, is mainly expressed in cardiac endothelial cells and becomes downregulated in mouse models of diabetes mellitus; (2) Methods: In this study, we investigated the impact of CTRP9 on early stages of diabetic cardiomyopathy induced by 12 weeks of high-fat diet; (3) Results: While the lack of CTRP9 in knock-out mice aggravated insulin resistance and triggered diastolic left ventricular dysfunction, AAV9-mediated cardiac CTRP9 overexpression ameliorated cardiomyopathy under these conditions. At this early disease state upon high-fat diet, no fibrosis, no oxidative damage and no lipid deposition were identified in the myocardium of any of the experimental groups. Mechanistically, we found that CTRP9 is required for insulin-dependent signaling, cardiac glucose uptake in vivo and oxidative energy production in cardiomyocytes. Extensive RNA sequencing from myocardial tissue of CTRP9-overexpressing and knock-out as well as respective control mice revealed that CTRP9 acts as an anti-inflammatory mediator in the myocardium. Hence, CTRP9 knock-out exerted more, while CTRP9-overexpressing mice showed less leukocytes accumulation in the heart during high-fat diet; (4) Conclusions: In summary, endothelial-derived CTRP9 plays a prominent paracrine role to protect against diabetic cardiomyopathy and might constitute a therapeutic target.

## 1. Introduction

Diabetes mellitus (DM) is a chronic disease resulting from the impaired production or inadequate effects of insulin. Recent estimates indicated that more than 450 million adults across the globe suffer from DM, and the numbers are projected to rise continuously [1]. The death toll associated with DM rose from less than one million in 2000 to 1.6 million people in 2016, whereby cardiovascular diseases (CVD) are the leading causes of mortality and morbidity in this patient population [2]. In 1972, the Framingham study detected a 5-fold increased risk of female and a 2.5-fold increased risk of male diabetes patients developing heart failure (HF) [3]. Cardiac dysfunction in the absence of coronary artery disease, valvular disease and hypertension is defined as diabetic cardiomyopathy [4]. Obesity, hyperglycemia and hyperinsulinemia were shown to promote the development of this disease. While the underlying mechanisms are increasingly being studied, knowledge about the molecules affecting cardiac insulin resistance or improving diabetic cardiomyopathy remains scarce.

In this study, we investigated the role of the secreted glycoprotein C1q and tumor necrosis factor related protein 9 (CTRP9) in the development of diabetic cardiomyopathy in mice. CTRP9 belongs to the highly conserved CTRP family of secreted proteins, of which 15 members (CTRP1-15) have so far been identified [5]. With the exception of CTRP4, all CTRPs consist of four domains: an N-terminal signal peptide, followed by a domain with one or more conserved Cys residues, a collagen-like domain with variable Gly-X-Y repeats, and a C-terminal globular C1q domain. While all CTRPs are structurally related to adiponectin, an insulin-sensitizing hormone secreted from adipose tissue, each CTRP appears to have its own unique expression profile and specific, non-redundant functions [5]. Of all CTRPs, CTRP9 has the highest structural similarity to adiponectin (amounting to 54% amino acid identity in the globular domain). CTRP9 is expressed highly in the heart and in smaller amounts in adipose tissue [6]. It is mainly secreted as trimer and can also be found in serum [7]. Within the heart, CTRP9 is predominantly expressed and secreted from cardiac endothelial cells, from where it acts on cardiomyocytes in a paracrine manner, but cardiomyocytes also express very small amounts of CTRP9 by themselves [6].

Recent studies suggest that the absence of CTRP9 in knock-out mice (KO) triggers obesity in advanced age, while systemic overexpression of CTRP9 might protect from weight gain and metabolic dysfunction [8,9]. With regard to the heart, administration of globular recombinant CTRP9 (lacking large parts of the N-terminal region) was protective by preventing cardiomyocyte death during ischemia/reperfusion injury in normal and diabetic mice, as well as in remodeling after myocardial infarction (MI) [10,11,12,13]. CTRP9 likely acts via the adiponectin receptor 1 (AdipoR1) and might mediate its protective effects by activating adenosine monophosphate (AMP)-dependent kinase, protein kinase B (AKT) or protein kinase A [10,11,12,14]. On the other hand, we demonstrated that CTRP9 plays a maladaptive role during cardiac pressure overload (by transverse aortic constriction, TAC), due to activation of the ERK5 (extracellular signal-regulated kinase 5)–GATA4 signaling axis in cardiomyocytes, triggering cardiac dysfunction, hypertrophy and fibrosis [6].

While CTRP9 expression increases in the heart of mice after TAC and in the cardiac tissue and serum of patients suffering from cardiac overload, myocardial levels and serum levels of CTRP9 decrease due to chronic high-fat diet (HFD) in diabetic mice [9,13]. This differential regulation of CTRP9 due to HFD might imply a special function of CTRP9 in the development of diabetic cardiomyopathy.

Here, we demonstrate that CTRP9 KO mice exert left ventricular diastolic dysfunction after 12 weeks of HFD feeding and that, on the other hand, overexpression of CTRP9 in the heart rescues HFD-induced diastolic dysfunction. Through dynamic ^18^F-fluoro-deoxyglucose (FDG) positron-emission tomography (PET) imaging and interrogation of cardiac signaling pathways, we reveal that CTRP9 is necessary to prevent cardiac insulin resistance and to mediate the myocardial uptake of glucose. By extensive RNA sequencing, we demonstrate an upregulation of genes related to activation of the immune response in CTRP9 KO mice, which is accompanied by an increased leukocyte count in the heart. Cardiac CTRP9 overexpression, in turn, counteracts inflammation and might be useful for ameliorating diabetic cardiomyopathy.

## 2. Materials and Methods

### 2.1. Animal Experiments

Mice were housed in groups of up to 5 animals with a 12:12 light cycle in the central animal facility of Hannover Medical School. CTRP9 knock-out mice were created as described previously [6]. Wild-type mice (WT), as well as homozygous (KO) and heterozygous (HETs) knock-out littermates were used in this study. Feeding of a high-fat diet (containing 60 kcal-% fat, #D12492 from Brogaarden, Lynge, Denmark) or control chow (#D12450J from Brogaarden) was started in 7–9 week-old male C57BL/6J mice (Charles River Laboratories, Wilmington, MA, USA), and was maintained for 12 weeks.

The open reading-frame of murine *C1qtnf9* (encoding for CTRP9) was cloned into the AAV-vector genome plasmid pdsTnT-Cre, and AAV9 was manufactured as previously described [6]. AAV9 vector expressing rluC (renilla luciferase) under the control of the same TnT promoter were used as control (AAV-Luc). An amount of 1 × 10^12^ vg of AAV9-control or AAV9-CTRP9 were injected intravenously into tail veins at the age of 7–9 weeks. After 12 weeks of diet, all mice were fasted for 6 h and sacrificed. All procedures involving the use and care of animals were performed according to the directive 2010/63/EU of the European Parliament and the council on the protection of animals used for scientific purposes. Approval was granted by the local state authorities (LAVES Niedersachsen, Oldenburg, Germany; No.: 33.12-42502-04-15/1953).

### 2.2. Echocardiography

A linear 30 MHz transducer (Vevo 3100, Visualsonics, Toronto, ON, Canada) and VevoLab software were utilized to measure End Diastolic Area, Fractional Area Change, Global Circumferential Strain, E’SAX-radial and Peak Diastolic Circumferential Strain Rate. Mice were anesthetized with 1–1.5% isoflurane, as previously described [15,16]. Fractional Area Change was calculated in short-axis B-mode images as ((left ventricular end-diastolic area − left ventricular end-systolic area)/left ventricular end-diastolic area) ×100 [15,16]. Diastolic and global cardiac function were analyzed post-processing using speckle-tracking algorithms within the VevoLab software and VevoStrain application, as reviewed in Ram et al. [17]; global circumferential strain was calculated as the percentage of increase of endocardial circumference from end-diastole to end-systole. As a marker of endocardial relaxation, the first peak of the average endocardial radial velocity curve during diastole was considered equivalent to the E′ wave as measured with tissue Doppler imaging. Additionally, peak diastolic circumferential strain rate was measured as derivative of strain over time (dStrain/dt). Here, the peak of the average circumferential strain rate curve during diastole was analyzed.

### 2.3. Serial ^18^F-FDG PET-CT Imaging

Non-fasted mice were anesthetized with 1.5% isoflurane at 0.6 L/min O_2_ and placed in a small animal PET camera (Inveon DPET, Siemens, Berlin, Germany), as previously described, while ECG and respiratory frequency were monitored [18]. All mice were injected with 6 mU/g body weight (BW) insulin (Human Insulin Normal 100, Lilly Pharma, Indianapolis, IN, USA) and 1 mg/g BW glucose i.p. 30 min prior to FDG injection to stimulate FDG uptake. Before and after the scans, blood glucose was measured at a peripheral vein using a glucose meter (Contour XT, Bayer Vital, Leverkusen, Germany). ^18^F-FDG (7.7 ± 0.9 MBq) was injected via a lateral tail vein catheter 20 min after the start of anesthesia. Dynamic images were acquired over 30 min. For anatomic colocalization, a low-dose micro-CT scan was acquired after the PET scan.

### 2.4. Image Processing and Analysis

List mode data of 30 min scans were histogrammed to 29 frames (5 × 2 s, 4 × 5 s, 3 × 10 s, 8 × 30 s, 5 × 60 s, 4 × 300 s). Images were reconstructed to a 128 × 128 × 159 image matrix (0.78 × 0.78 × 0.80 mm) using an ordered subset expectation maximization three-dimensional/maximum a posteriori (OSEM3D/fastMAP) algorithm (β = 0.01, OSEM subsets = 16, OSEM iterations = 2, MAP iterations = 18, constant resolution prior) with scanner-applied scatter correction and a target resolution of 1.2 mm. An external source transmission scan was obtained for each mouse and used for attenuation correction (paired 185 MBq ^57^Co sources, 10 min acquisition) as previously described [18]. Images were analyzed using PMOD 3.6 software (PMOD Technologies, Fallanden, Switzerland).

Rate of myocardial glucose uptake (rMGU) was calculated as K_i_ × (BG/LC); where K_i_ is the graphically-defined Patlak slope, BG is the average blood glucose concentration of start and end of the scan, and LC is the lumped constant = 0.67 estimated for rodents. Patlak values were compared to sequential semi-quantitative myocardial ^18^F-FDG uptake (percent injected dose per gram, %ID/g BW), as described in Thackeray et al. [19].

### 2.5. Serum Analysis

Mice were fasted for 6 hours. Blood Glucose was measured after micro puncture of a tail vein with Accu-Check Aviva (Roche, Basel, Switzerland).

The glucose challenge test was conducted after mice were fasted for 6 h. They were injected with 1 mg/g/BW Glucose (G20, Braun, Melsungen, Germany). Blood glucose levels were measured after 5, 10, 15, 30, 60 and 120 min. The HOMA Index of Insulin Resistance was calculated as (fasting serum insulin (ng/mL) × fasting serum glucose (mM))/22.5.

For the detection of insulin in murine serum, the Ultra-Sensitive Mouse Insulin ELISA Kit was used (#90080, Crystal Chem, Elk Grove Village, IL, USA), following the protocol as suggested by the manufacturer.

### 2.6. Isolation of Adult Cardiomyocytes

Isolation of adult cardiac myocytes was achieved by using a Langendorff perfusion system, as described [6,20]. Briefly, immediately after excision, the heart was placed into ice-cold 1× perfusion buffer (10x stock: 1130 mM NaCl, 47 mM KCl, 6 mM KH_2_PO_4_, 6 mM Na_2_HPO_4_, 12 mM MgSO_4_, 120 mM NaHCO_3_, 100 mM KHCO_3_, 100 mM HEPES buffer solution and 300 mM Taurine) to clean the tissue around the aorta under a microscope. The aorta was then placed onto a cannula (inner diameter 1 mm) with forceps and tied with silk suture to allow perfusion through the coronary arteries. Subsequently, the heart was perfused with enzyme solution containing Liberase DH (5 mg/mL, Roche, 5,401,089,001), trypsin (1%, Gibco, Waltham, MA, USA, 15,090,046) and CaCl_2_ (100 mM) to digest the tissue at 37 °C. Enzymatic digestion was ended with Stop I and II solutions containing perfusion buffer with FCS and CaCl_2_ (10 mM). Digested heart tissue samples were passed through a 100 µm cell strainer. Calcium concentration was then gradually increased by manual administration into digested heart tissue with continuous gently mixing.

### 2.7. Seahorse Extracellular Flux Analysis

Metabolic analysis was measured with Seahorse XFe24 analyzer using Seahorse XF substrate oxidation stress kits for glucose/pyruvate (Agilent, 103,673-100). To understand the changes in cardiomyocyte metabolism, adult WT and CTRP9-KO cardiomyocytes were isolated with Langendorff perfusion; 12,000 cardiomyocytes were then plated on Seahorse 24-well FluxPak plates and cultured for 24 h. On the next day, culture medium of the cells was exchanged with Seahorse XF DMEM (Agilent, 103,575-100), including 1 M glucose (Agilent, 103,577-100), 100 mM pyruvate (Agilent, 103,578-100) and 200 mM glutamine (Agilent, 103,579-100) and incubated in a non-CO_2_ incubator at 37 °C for 1 h. For experimental groups UK5099 (2 μM) inhibitor was loaded into the cartilage, while the control groups were treated with assay medium alone. Previously optimized amounts of oligomycin, FCCP and rotenone/antimycin A were also loaded into the cartilage. Substrate stress-test assays were run according to the manufacturer’s protocol and analyzed using Seahorse Analytics web-based software. Data were normalized to the number of cells, which was assessed with colorimetric measurement after staining with crystal violet assay kit (Abcam, Cambridge, UK, ab232855).

### 2.8. Western Blot Analysis

Western blot analyses were performed using standard procedures using anti-P-AKT (T308) (#2965), anti-P-AKT (S473) (#9271), anti-AKT (#9272), anti-P-S6 (S240/244) (#5364), anti P-S6 (S235/236) (#4857), anti-S6 (#2217), anti-P-GSK (#9336), anti-GSK (#9322), anti-P-ERK 1/2 (#9101), anti-ERK 1/2 (#9102) (all Cell Signaling), anti GAPDH (#10R-G109a, Fitzgerald, North Acton, MA, USA) and anti-CTRP9 (#mab6537, R & D Systems, Minneapolis, MN, USA). Quantification was performed using ImageJ software version 1.52a (NIH, Bethesda, MD, USA).

### 2.9. Histology

After removal from the chest cavity, the hearts were rinsed in PBS and 0.5 M KCl. Transversal frozen sections (7 µM in thickness) of the myocardium were generated. To measure cardiomyocyte dimensions, the cardiomyocyte cell membranes were stained with tetramethyl rhodamine isothiocyanate-conjugated wheat-germ agglutinin (TRITC-WGA, Sigma Aldrich, Taufkirchen, Germany) and the nuclei with 4′,6-diamidino-2-phenylindole (DAPI, Vector Laboratories, Newark, CA, USA). Analysis was conducted with ImageJ software version 1.52a (NIH, Bethesda, MD, USA).. For the detection of fibrosis or lipid accumulation, OCT-embedded hearts were sectioned at 12 µm thickness and stained with Sirius red or Oil Red O to detect collagen fibers or lipid droplets, respectively. The quantification of myocardial fibrosis was conducted with Photoshop CC 2018 (Adobe, San Jose, CA, USA). For immunofluorescence imaging, slides of 7 µm were fixed with formalin, followed by blocking with 3% BSA solution, before overnight incubation with primary antibody and incubation for 2 h with secondary antibody. WGA and DAPI were used as counterstains, as described above. The following antibodies and dilutions were used: rat anti-CD45 (#550539, BD Biosciences, Franklin Lakes, NJ, USA, 1:50), rabbit anti-4HNE (bs-6313R, Bioss Antibodies, Woburn, MA, USA, 1:50), goat anti-rabbit AF488 (#4412, 1:200) and goat anti-rat AF488 (#4416, both Cell Signaling, 1:200)

### 2.10. RNA Sequencing and Bioinformatics

RNA isolation from tissue samples was performed with QIAzol reagent (Qiagen, Hilden, Germay, 79,306), and from isolated cells with NucleoSpin RNA isolation kit (Macherey-Nagel, Duren, Germany, 740,955.250) according to the manufacturers’ protocols. RNA quality and integrity scores (Agilent 21,000 Bioanalyser, Santa Clara, CA, USA), library preparation (DNBSEQ Eukaryotic Strand-specific mRNA library) and sequencing (Illumina HiSeq 2500) were performed at BGI, Hong Kong. Bulk RNA sequencing was performed as single-ended 50-base sequence read length.

Before alignment to the reference genome (mm^10^), the reads were trimmed with the *FastqCleaner* R package to remove the adapter sequences. Trimmed reads were then aligned to the reference genome by the *bowtie2* alignment tool, and gene annotation was performed with the *bioMaRt* R package. Library size was normalized to read counts per million (cpm), and log_2_ values were calculated for further analysis of differential gene expression using the *edgeR* package. Genes were filtered with *p*-value < 0.05 for significant changes. Heatmaps showing the differentially expressed genes were generated using the *heatmap.2* function in the ggplot2 library in R. Gene ontology analyses were performed with Metascape online tool separately for all significantly (*p* < 0.05) up- and downregulated mRNAs. RNAseq data sets were deposited in the National Center for Biotechnology Information’s Gene Expression Omnibus database, with accession number GSE219101.

### 2.11. Statistics

Data analysis and statistical analysis were performed with Prism software Version 6 (GraphPad, Boston, MA, USA. Data are shown as mean ± standard error of the mean (SEM). All the experiments were carried out in at least three biological replicates. The investigators were blinded for mouse genotype and treatment during echocardiography, organ weight determination and all histological and immunofluorescence quantifications. An unpaired two-tailed Student *t*-test was used for comparing two groups. Multiple groups were compared with one-way ANOVA and Fisher’s LSD post-hoc test. Values of *p* < 0.05 were considered statistically significant.

## 3. Results

### 3.1. High-Fat Diet Leads to Peripheral Insulin Resistance in CTRP9 KO Mice

In order to examine the contribution of endogenous CTRP9 to the development of diabetic cardiomyopathy, eight-week-old wild-type (WT), as well as homozygous (KO) and heterozygous (HET) CTRP9 knock-out mice were fed either a diet containing 60 kcal % fatty acids (HFD, as described [21]) or a standardized control chow for 12 weeks. The experimental time schedule is shown in Figure 1A. The HFD led to a significant increase in body weight vs. chow after 12 weeks, but no differences were detected between WT, HET and KO mice (Figure 1B). Fasted serum glucose levels were significantly elevated in HFD-fed KOs compared to WT mice (Figure 1C). A glucose challenge revealed strongly elevated glucose levels and significantly delayed normalization of serum glucose in HFD-fed KO vs. WT mice (Figure 1D). Moreover, serum insulin levels were significantly elevated in HFD-fed KO mice (Figure 1E). Together with the elevated glucose levels, this led to an increased HOMA index (directly depending on serum glucose and insulin levels under fasting conditions), which indicated a state of insulin resistance in KO mice after HFD feeding (Figure 1F).

### 3.2. Reduced CTRP9 Levels Trigger Impaired Diastolic Heart Function during HFD

We next investigated the functional consequence of reduced CTRP9 levels for the myocardium during HFD feeding. The heart weight/tibia length ratio (HW/TL) increased significantly in KO mice receiving HFD vs. control chow, but this was neither observed in WT nor in HET mice (Figure 2A). The lung weight/TL ratio (LW/TL) was not changed between the groups (Figure 2B). Echocardiographic analysis showed no differences in cardiac enddiastolic area (ENDOarea) or systolic heart function (fractional area change, FAC) between genotypes (Figure 2C,D). It was previously demonstrated that diabetes leads mainly to diastolic dysfunction^14^. Interestingly, global circumferential strain (GCS) as a marker for global cardiac function, cardiac relaxation (E′_SAX-radial_) and peak diastolic strain rate (as a marker for lusitropy) indicated significantly decreased diastolic function in KO (and partially also in HET) compared to WT mice on HFD (Figure 2E–G).

### 3.3. CTRP9 Knockout Leads to Impaired Cardiac Glucose Utilization In Vivo

Because altered cardiac glucose uptake had been implicated in the development of diabetic cardiomyopathy, we conducted dynamic ^18^F-fluoro-deoxyglucose (FDG) positron-emission tomography (PET) (Figure 3A). Insulin-stimulated cardiac FDG uptake was increased in HFD-fed WT mice compared to mice receiving control chow. This increase in glucose uptake, however, was markedly blunted in CTRP9 KO mice, which exerted an impaired rate of myocardial glucose uptake (rMGU) in the heart compared to WTs during HFD (Figure 3B,C), suggesting that CTRP9 plays a key role in mediating increased cardiac glucose uptake under these conditions.

We next interrogated whether defects in insulin-dependent signaling might underlie the decreased insulin stimulated glucose uptake in the hearts of KO mice. Therefore, we injected insulin 10 min before sacrifice and compared the phosphorylation of insulin-dependent signaling proteins in the myocardium with PBS-injected controls (Figure 3D and Appendix A). Knockout of CTRP9 led to decreased phosphorylation of AKT, ribosomal S6 protein and Glycogen Synthase Kinase 3β (GSK3β) after insulin injection, but showed no differences in the phosphorylation of ERK1/2 MAP kinases. Therefore, we found decreased responsiveness to insulin (i.e., insulin resistance) in the myocardium of KO mice.

In order to assess cardiomyocyte metabolism, we conducted a Seahorse analysis (Figure 3E,F). Interestingly, adult cardiomyocytes of CTRP9 KO mice exerted a markedly decreased maximum mitochondrial respiration rate compared to WT cells, which was not further decreased by inhibition of the mitochondrial pyruvate carrier by the specific inhibitor UK5099. WT mice, in contrast, showed a strong reduction of mitochondrial respiration during UK5099 treatment, indicating their strong dependence on glucose metabolism (Figure 3E,F).

### 3.4. Cardiac Overexpression of CTRP9 Leads to Impaired Peripheral Glucose Sensitivity

Because CTRP9 deficiency triggered diabetic cardiomyopathy during HFD feeding, we next wanted to test whether enhancement of CTRP9 signaling ameliorated the disease. We therefore overexpressed CTRP9 in cardiomyocytes by intravenous administration of AAV9-(TropT)-CTRP9, which led to the solid overexpression of CTRP9, specifically in the heart due to the Troponin T promoter elements included in the vector [22]. AAV9-(TropT)-Luciferase was used as a control (Figure 4A,B). Mice were randomized to receive either AAV-CTRP9 or AAV-Luc, following the experimental scheme visualized in Figure 4A. AAV-control as well as AAV-CTRP9-injected mice similarly gained body weight in response to HFD, although at the end of the experiment, AAV-CTRP9-treated mice were significantly heavier than AAV-Luc treated mice (Figure 4C). Chow-fed mice showed no differences in body weight in response to any AAV treatment. Interestingly, AAV-CTRP9-treated mice, but not AAV-Luc-treated mice on HFD displayed significantly elevated fasting glucose levels and higher glucose levels as well as their delayed normalization during a glucose challenge (Figure 4D,E). AAV-CTRP9-treated mice also exerted increased serum insulin levels and an elevated HOMA-Index, indicating peripheral insulin resistance in response to cardiac CTRP9 overexpression during HFD feeding. AAV-control injection did not lead to increased insulin levels or an elevated HOMA-Index (Figure 4F,G).

### 3.5. Overexpression of CTRP9 Improved Diastolic Cardiac Function and Increased Cardiac Glucose Metabolism

We next deciphered the impact of CTRP9 over-expression during HFD on heart growth and cardiac function. While the HW/TL ratio slightly, but not significantly, increased in AAV-Luc mice during HFD, no significant increase occurred in mice with CTRP9 overexpression (Figure 5A). Similarly, we observed an increase in LW/TL ratio in AAV-Luc, but not in AAV-CTRP9-treated mice on HFD, indicating less pulmonary congestion due to better cardiac function in the CTRP9-overexpressing mice (Figure 5B). Echocardiography did not reveal changes in LVEDA or FAC between groups (Figure 5C,D), but global circumferential strain, peak diastolic circumferential strain rate and E′_SAX-radial_ indicated improved diastolic function in AAV-CTRP9 vs. AAV-Luc-treated mice during HFD (Figure 5E–G).

To investigate the impact of CTRP9 overexpression on cardiac glucose uptake, we conducted dynamic FDG PET scans, which showed a trend towards increased FDG uptake in the heart of AAV-CTRP9 compared to AAV control during HFD (Figure 6A,B). Furthermore, we analyzed insulin-dependent signaling pathways in the hearts by immunoblotting. In this analysis, no major differences were found between AAV-CTRP9- and AAV-Luc-treated mice, with or without insulin treatment (Figure 6C and Appendix A).

### 3.6. Histological Examination Revealed Increased Cardiomyocyte Hypertrophy upon CTRP9 Knock-Out and CTRP9 Overexpression

Histological examinations revealed an increased cardiomyocyte cross-sectional area in CTRP9 KO mice vs. WT and HET mice during HFD exposure. No increase in cardiomyocyte size was observed in WT mice during HFD (Figure 7A). The capillary density did not change in HFD in KO, WT or HET mice (Figure 7B). Notably, cardiomyocyte hypertrophy also developed due to AAV-CTRP9 treatment during HFD administration, but not in the AAV-Luc treatment (Figure 7C and Appendix A). Capillary density did not change under these conditions (Figure 7D). In addition, analysis of myocardial fibrosis by Sirius red staining, of lipid deposition by oil red o staining and of oxidative damage by staining for 4-hydroynonenal (4-HNE) did not reveal any differences between WT, KO and HET or AVV-Luc and AAV-CTRP9 treatment during chow or HFD feeding (Figure 7E,F and Appendix A).

### 3.7. RNA Sequencing Reveals Anti-Inflammatory Effects of CTRP9 Overexpression during HFD Administration

Comparison of gene expression between HFD-fed WT and KO mice revealed the upregulation of genes related to (gene-ontology classes, biological processes) the tricarboxylic acid (TCA) metabolic process and the nucleoside triphosphate metabolic process, including several mitochondrial proteins belonging to the tricarboxylic acid cycle or the respiratory chain (Figure 8A). Example genes are indicated in the Figure. In addition, immune response and ribonucleoprotein complex biogenesis genes were upregulated in the myocardium of KO vs. WT mice. Genes related to cardiomyocyte differentiation, the response to steroid hormones, RNA splicing and the regulation of the membrane potential were downregulated in KO vs. WT mice during HFD treatment. Altered splicing behavior was detected in the myocardium of CTRP9 KO in HFD treatment, whereby we mainly observed increased exon skipping events in KO vs. WT mice (Appendix A). As an example, differential splicing with increased exon skipping of Kat5 (a histone acetylase) and Mef2d (a pro-hypertrophic transcription factor) in HFD-fed KO mice are displayed (Appendix A). In general, mainly genes related to mitochondrial transport, autophagy, mitochondrial organization and cellular catabolic and metabolic processes displayed exon skipping events during HFD treatment (Appendix A).

The comparison of myocardial RNA sequencing results of chow- vs. HFD-treated WT mice and chow- vs. HFD-treated KO mice are displayed in Appendix A. It is demonstrated there that in response to HFD (vs. chow), WT mice induce genes related to extracellular matrix, lipid metabolism, cell migration and cellular response to growth factors, while downregulating TCA-cycle genes and muscle genes. KO mice also induce lipid and fatty acid metabolic, angiogenesis and cell migration genes, but in contrast induce chemotaxis genes. KO mice also downregulate muscle genes, but specifically reduce carbohydrate and amino acid metabolism gene.

RNA sequencing of mouse hearts treated with AAV during HFD feeding revealed that cardiac CTRP9 overexpression led to upregulation of the genes positively regulating cholesterol flux, intracellular protein transport and endocytosis. On the other hand, genes related to cytokine production, inflammation, extracellular matrix organization and possible regulation of cell death genes were downregulated upon CTRP9 overexpression in the heart during HFD administration (Figure 8B).

Because inflammatory gene expression was upregulated in KO mice and downregulated upon AAV-mediated CTRP9 overexpression in HFD feeding, we assessed myocardial inflammation by immunostaining for CD45 (a leukocyte marker). Indeed, as displayed in Figure 8C, HFD increased leukocyte abundance in the heart of control mice (WT mice with or without AAV-Luc administration), which was aggravated by the lack of CTRP9 in KO mice but ameliorated by AAV9-mediated CTRP9 overexpression. Hence, CTRP9 exerts anti-inflammatory effects in diabetic cardiomyopathy.

## 4. Discussion

Cardiac dysfunction in diabetic cardiomyopathy is based on multiple mechanisms. For example, the diabetic heart exerts increased fatty acid oxidation, but reduced oxidation of glucose [23,24]. This is believed to be the consequence of (a) reduced glucose uptake in insulin-resistant states, and (b) increased cellular fatty acid uptake through the membrane transporter Cd36 into cardiomyocytes. As ATP production from fatty acids requires more oxygen than from glucose, the heart becomes more inefficient. More reactive oxygen species (ROS) are produced, which foster mitochondrial uncoupling and therefore a further reduction in mitochondrial energy production. In addition, toxic lipid metabolites trigger ER stress, myocyte death and dysfunction. Increased levels of pro-inflammatory cytokines and increased leukocyte infiltrations are also observed in diabetic hearts [23,24].

In this study, we investigated how CTRP9 affects cardiac glucose uptake, global myocardial gene expression and cardiac histology and function during HFD-induced DM. CTRP9 is highly expressed in the heart (about 100-fold higher than the related adiponectin) and is mainly secreted from endothelial cells to act in a paracrine manner [6]. CTRP9 is upregulated in cardiac pressure overload, when it contributes to cardiac hypertrophy and dysfunction [6]. Interestingly, however, during myocardial ischemia, endogenous CTRP9 expression is downregulated, and supplementation of exogenous CTRP9 promotes cell survival and improves ventricular remodeling [12,13]. Because cardiac as well as plasma CTRP9 levels markedly decrease during HFD, the question of whether or not reduced CTRP9 levels contribute to myocardial dysfunction under these circumstances has a high clinical relevance [13]. Recently, CTRP9 KO mice were analyzed after long-term (26 weeks) exposure to HFD, which resulted in advanced disease, i.e., more systolic cardiac dysfunction, enhanced oxidative damage, lipid deposition and fibrosis in KO mice, although neither CTRP9 overexpression nor cardiac metabolism were investigated, and no underlying molecular mechanisms were provided [25]. In our study, we only administered HFD for 12 weeks and thereby investigated an early disease state, to gain knowledge regarding the molecular basis of the CTRP9 effects. At the end of HFD administration, both WT and KO mice exerted a reduced glucose tolerance, with a significant worsening in KO mice, but insulin resistance was only observed in KO mice, as previously published [8]. While HFD did not affect cardiac function in WT mice at this stage, it triggered diastolic left ventricular dysfunction in KO mice, which is known to occur before systolic dysfunction in diabetes [26]. At the same time, only a mild increase in cardiomyocyte hypertrophy, but no enhancement of fibrosis, no lipid deposition and no oxidative damages, were observed in KO mice. Therefore, although these features might occur at the advanced disease state, they are likely not primarily responsible for aggravated cardiomyopathy in CTRP9 KO mice. Highly selective overexpression of CTRP9 in cardiomyocytes, in turn, improved diastolic function.

Unexpectedly, however, we observed that cardiac CTRP9 overexpression induced a state of systemic insulin resistance with increased serum insulin levels. This is in contrast to systemic transgenic CTRP9 overexpression in mice, which protected from diet-induced obesity and metabolic dysfunction by triggering reduced food intake, an increased oxygen consumption rate in part due to enhanced mitochondrial content, increased fatty acid oxidation enzyme expression and chronic AMPK activation in skeletal muscle [9]. In these transgenic mice, CTRP9 was overexpressed in heart, skeletal muscle and the brain. Therefore, cardiac CTRP9 overexpression in our study might not be sufficient to alleviate systemic metabolic dysfunction, because skeletal muscles especially serve this purpose. In addition, the overexpression in cardiomyocytes in our study is, to a certain degree, unphysiological, because CTRP9 is mainly derived from endothelial cells in the heart. However, due to the secreted nature of the protein, the originating cell might be less relevant. Why the metabolic state even worsens due to cardiac CTRP9 overexpression currently remains unresolved, although the counter regulatory effects of systemic endocrine systems due to altered cardiac metabolism could play a role. In this regard, it had been demonstrated that the heart can regulate metabolic activity and energy expenditure in peripheral tissues, for example, by releasing endocrine (“cardiocrine”) factors, such as natriuretic peptides or MG53 [27,28,29]. Therefore, although we have not investigated this, an improved cardiac insulin responsiveness due to selective myocardial CTRP9 overexpression could modify cardiocrine signaling in a way that leads to insulin resistance in peripheral tissues and increased systemic insulin levels.

We found in our study that lack of CTRP9 during HFD in KO mice blunted insulin-triggered signaling and glucose uptake in the heart in vivo. This means that reduced CTRP9 levels in the myocardium in diabetic hearts promote cardiac insulin resistance, which typically leads to an overall decrease in glycolysis and glucose oxidation and an increased use of fatty acids as energy source. This altered myocardial substrate use increases the oxygen cost of contractility and makes the heart more inefficient [30,31]. In turn, CTRP9 overexpression enhanced insulin-triggered cardiac glucose uptake, although no augmentation of signaling was seen. CTRP9 is also known to increase mitochondrial numbers in its target cells [9,32]: Both effects, i.e., decreased glucose uptake and reduced mitochondrial number, will likely contribute to reduced maximum mitochondrial respiration in the isolated cardiomyocytes of KO mice that we detected by Seahorse analysis. Moreover, mitochondrial dysfunction emerges as a consequence of enhanced fatty acid metabolism and resulting ROS production and lipotoxicity [33,34,35]. Intriguingly, while inhibition of the mitochondrial pyruvate carrier strongly reduced maximum mitochondrial respiration in WT cardiomyocytes roughly to the level of KO cardiomyocytes, no further reduction was induced by pyruvate carrier inhibition in the latter, because apparently pyruvate usage was already minimal due to reduced glucose uptake and metabolism. It is interesting to note that endothelial cells appear to play an important regulatory function for the metabolism of cardiomyocytes. While previous work had revealed their essential contribution for the uptake of fatty acids [36], we suggest here that paracrine endothelial CTRP9 is necessary for adequate glucose uptake in cardiomyocytes. In summary, we propose that CTRP9 deficiency entails myocardial inefficiency and energy shortage and thereby leads to cardiac dysfunction.

To obtain further insight into the mechanisms of CTRP9 in diabetic hearts, we conducted extensive RNA sequencing analyses from mouse heart tissue. These experiments, in part, confirmed the effects from our metabolic analyses; for example, the reduced expression of glucose metabolism genes and highly significant upregulation of fatty acid metabolism genes in KO mice upon HFD. On the other hand, novel aspects were uncovered, such as reduced expression of cardiomyocyte-specific genes, such as Actc1 or Gata4, or the reduced expression of splicing factors in the myocardium of CTRP9 KO mice that result mainly in exon skipping in genes related to mitochondria, autophagy and metabolism and might therefore contribute to the accelerated development of diabetic cardiomyopathy. Indeed, changes in RNA splicing patterns had been previously found in diabetic cardiomyopathy and were suggested to promote the disease [37,38]. Moreover, the modestly increased expression of TCA cycle and respiratory chain genes in the myocardium of KO mice might be based on a mechanism to compensate for mitochondrial and metabolic dysfunction. Because immune response genes were upregulated in the myocardium of KO mice, but reduced upon CTRP9 overexpression under HFD conditions, we investigated this further and indeed found that CTRP9 inhibits leukocyte abundance in the diabetic heart. Although we have not further characterized the type of inflammatory cell that is suppressed by CTRP9, previous studies in diabetic cardiomyopathy have mainly implicated macrophages and T-cells [39]. For instance, pro-inflammatory M1 macrophages were reported to be more abundant and contribute to cardiac damage in diabetic heart failure [40,41,42]. Similarly, CD4-positive Th1 cells are especially upregulated in the peripheral blood and hearts of diabetic patients and might contribute to cardiac damage [43,44,45]. In addition, neutrophils and B-cells could play a role, but more work is needed to define the role of specific immune cell subsets in diabetic cardiomyopathy [39].

Because different anti-inflammatory maneuvers improved the experimental diabetic cardiomyopathy in mice and rats, the anti-inflammatory effects of CTRP9, which had been previously observed in vitro and were also described for the related adiponectin, likely contribute to the protective action of CTRP9 in diabetic cardiomyopathy [46,47,48,49,50,51,52]. As limitations of our study, we (a) did not decipher whether or not the protective effects of CTRP9 in diabetic cardiomyopathy depended primarily on its impact on cardiomyocyte metabolism or on its anti-inflammatory nature,(b) we only investigated the effects of CTRP9 in one model of diabetic cardiomyopathy and (c) only in male mice. Future studies should therefore investigate the effects of CTRP9 in additional disease models of DM (e.g., in db/db mice), and in female mice, as well as address the primary beneficial target of CTRP9 more in depth.

## 5. Conclusions

In summary, we demonstrate here that endothelial-derived CTRP9 is required to counteract insulin resistance and myocardial inflammation and to promote mitochondrial metabolism in order to protect from diabetic cardiomyopathy during HFD conditions. We envisage that restoration of CTRP9 levels in diabetic cardiomyopathy might be a novel therapeutic strategy, but care must be taken that systemic insulin resistance is not aggravated by this approach. To avoid this, systemic rather than targeted cardiac CTRP9 overexpression might be advisable.

## Figures and Tables

**Figure 1 cells-12-00443-f001:**
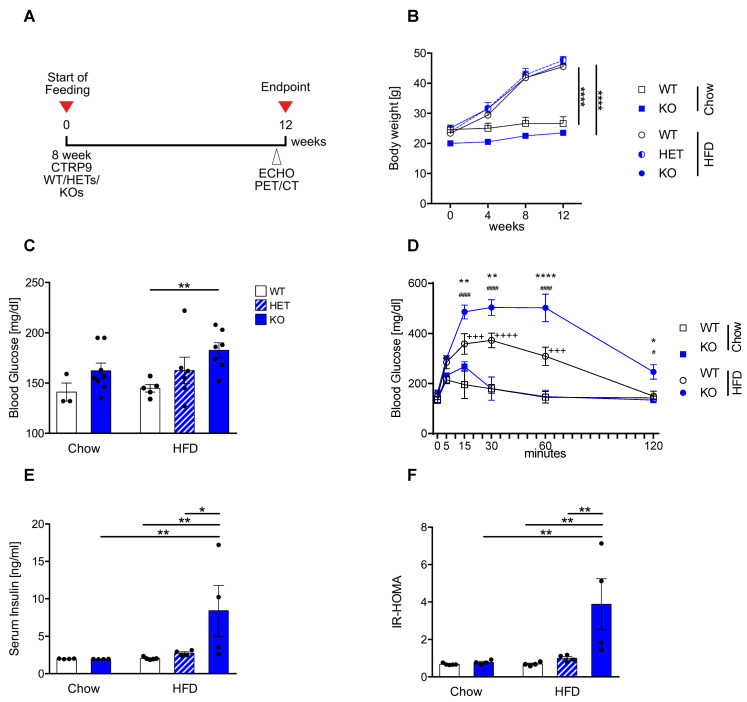
CTRP9 knock-out leads to peripheral insulin resistance after high-fat diet (HFD) feeding. (**A**) Scheme showing the experimental timeline. (**B**) Body weight measured 4, 8 and 12 weeks after start of diet. WT: wild-type mice; KO homozygous and HET heterozygous CTRP9 knock-out mice. (**C**) Fasted serum glucose levels. (**D**) Glucose tolerance test. */**/**** indicates significance between WT HFD vs. KO HFD. ^+++^/^++++^ indicates significance between WT HFD vs. WT Chow. ^#^/^####^ indicates significance between KO HFD vs. KO Chow. (**E**) Serum insulin levels. (**F**) HOMA-Index of insulin resistance. Data are shown as mean ± standard error of the mean (SEM). * *p* < 0.05, ** *p* < 0.01, **** *p* < 0.0001, # *p* < 0.05, ^####^
*p* < 0.0001, ^+++^
*p* < 0.001, ^++++^
*p* < 0.0001.

**Figure 2 cells-12-00443-f002:**
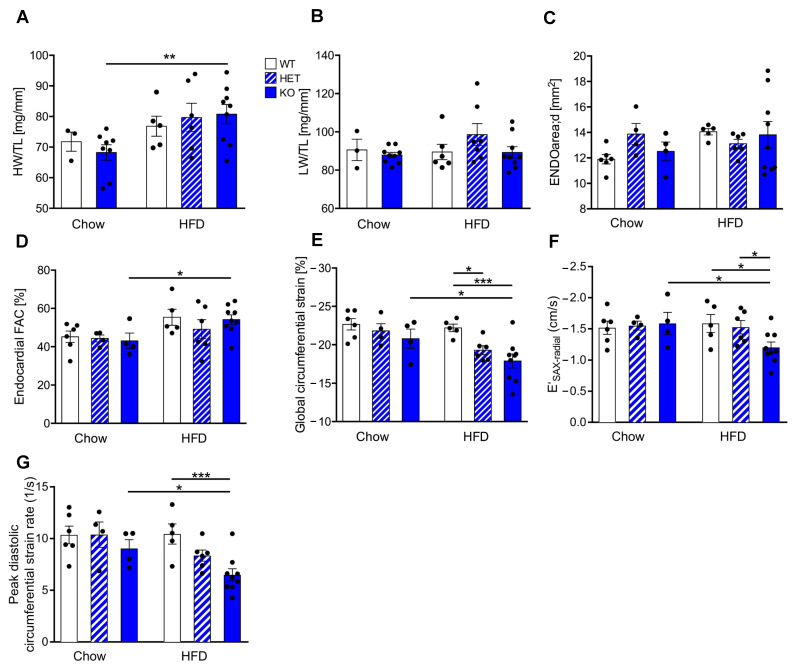
CTRP9 knock-out leads to increased heart weight and impaired cardiac diastolic function during high-fat diet (HFD). (**A**) Ratio between heart weight (HW) and tibia length (TL). WT: wild-type mice; KO homozygous and HET heterozygous CTRP9 knock-out mice. (**B**) Ratio between lung weight (LW) and TL. (**C**,**D**) Echocardiographic analysis. (**C**) Enddiastolic area. (**D**) Endocardial fractional area change (FAC). (**E**) Global circumferential strain as marker of global cardiac function. (**F**) E′_SAX-radial_ indicates average endocardial movement during cardiac relaxation. (**G**) Peak diastolic circumferential strain rate as a marker of lusitropy. Data are shown as mean ± standard error of the mean (SEM). * *p* < 0.05, ** *p* < 0.01, *** *p* < 0.001.

**Figure 3 cells-12-00443-f003:**
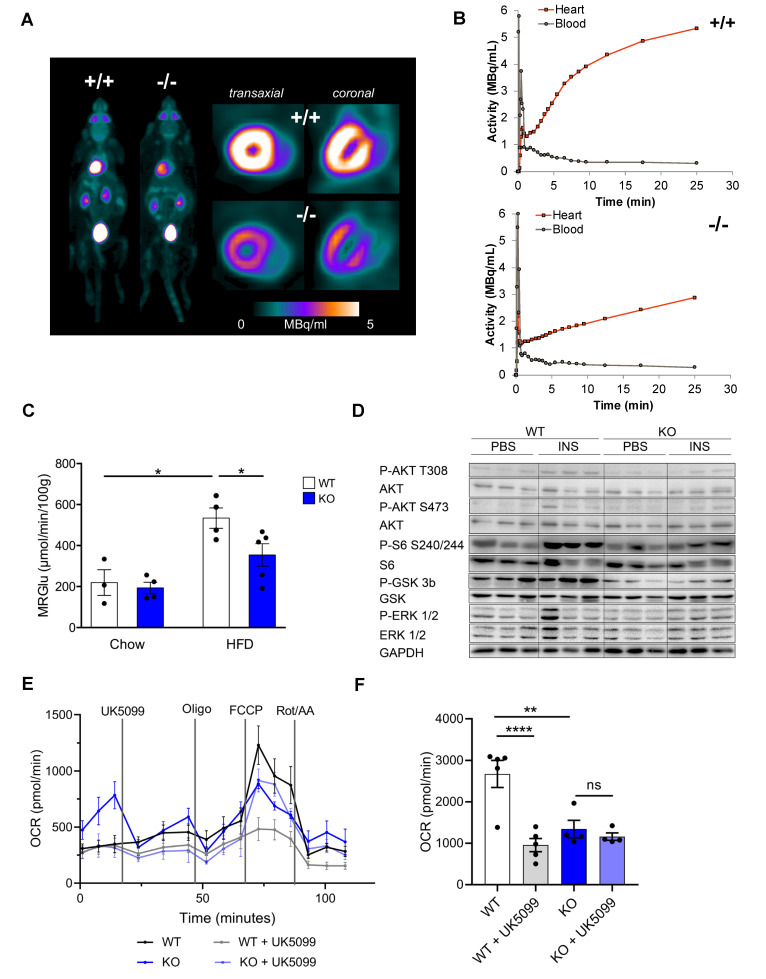
Decreased cardiomyocyte oxidative metabolism and reduced cardiac insulin response in CTRP9 knock-out mice during high-fat diet (HFD). (**A**) Representative images of whole mouse and cardiac PET scans of HFD-fed mice of the indicated genotypes. ^18^F-fluoro-deoxyglucose was used as a tracer. (**B**) Dynamic glucose uptake measured through PET scans, and its quantification in (**C**). WT: wild-type mice (+/+); KO: homozygous CTRP9 knock-out mice (−/−). (**D**) Immunoblot of cardiac protein lysate of mice of the indicated genotype and treated as shown. GAPDH was used as a loading control. INS: insulin. (**E**,**F**) Seahorse analysis of cardiomyocytes isolated from the indicated mice and treated as shown during the analysis. (**E**) Kinetic graph of the glucose/pyruvate stress test assay, and (**F**) bar chart of maximum respiration with or without inhibition of the mitochondrial pyruvate carrier by UK5099. Oligo: Oligomycin, Rot/AA: Rotenone/antimycin A. Data are shown as mean ± standard error of the mean (SEM). * *p* < 0.05, ** *p* < 0.01, **** *p* < 0.0001.

**Figure 4 cells-12-00443-f004:**
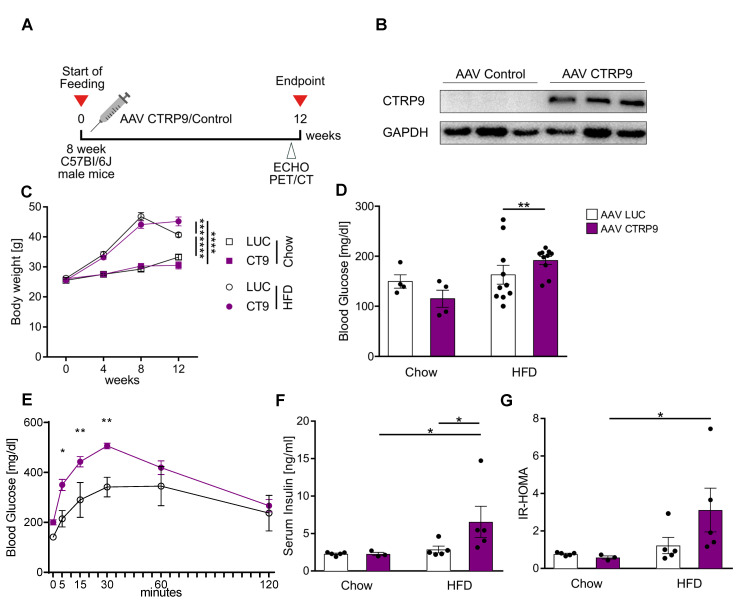
Cardiac overexpression of CTRP9 leads to peripheral insulin resistance during administration of a high-fat diet (HFD). (**A**) Scheme showing the experimental timeline. (**B**) Immunoblotting for CTRP9 from cardiac protein lysate of mice treated either with AAV Luc (control, LUC) or AAV CTRP9 (CT9, leading to myocardial CTRP9 overexpression). (**C**) Body weight measured at 4, 8 and 12 weeks in the indicated mice after start of feeding. (**D**) Fasted serum glucose levels. (**E**) Glucose tolerance test in the indicated mice after 12 weeks of HFD. (**F**) Serum insulin levels. (**G**) HOMA-Index of insulin resistance. Data are shown as mean ± standard error of the mean (SEM). * *p* < 0.05, ** *p* < 0.01, *** *p* < 0.001, **** *p* < 0.0001.

**Figure 5 cells-12-00443-f005:**
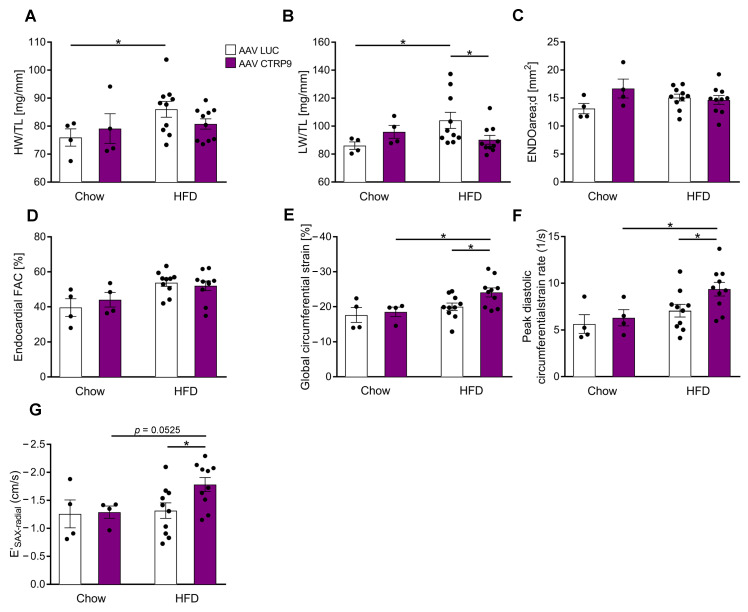
Overexpression of CTRP9 improves cardiac diastolic function during high-fat diet (HFD). (**A**) Ratio between heart weight (HW) and tibia length (TL) in the indicated mice treated either with AAV Luc (control) or AAV CTRP9 (leading to myocardial CTRP9 overexpression). (**B**) Ratio between lung weight (LW) and TL. (**C**–**G**) Echocardiographic analysis. (**C**) Enddiastolic area. (**D**) Endocardial fractional area change (FAC). (**E**) Global circumferential strain as a marker of global cardiac function. (**F**) Peak diastolic circumferential strain rate as a marker of lusitropy. (**G**) E′_SAX-radial_ based on average endocardial movement during cardiac relaxation. Data are shown as mean ± standard error of the mean (SEM). * *p* < 0.05.

**Figure 6 cells-12-00443-f006:**
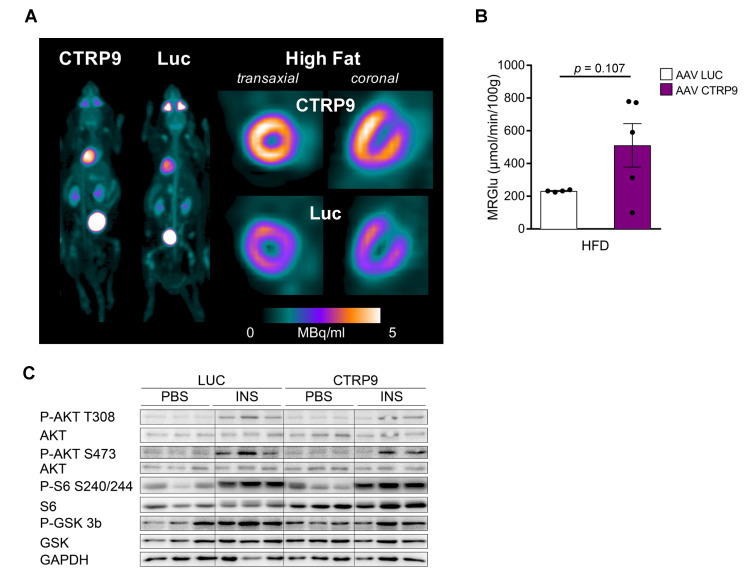
Overexpression of CTRP9 leads to increased glucose uptake in the heart during high-fat diet (HFD) administration. (**A**) Representative images of cardiac PET scans of the indicated HFD-fed mice treated either with AAV Luc (control) or AAV CTRP9 (leading to myocardial CTRP9 overexpression). ^18^F-fluoro-deoxyglucose was used as a tracer. (**B**) Dynamic glucose uptake measured through PET scans as described in (**A**). (**C**) Immunoblot of cardiac protein lysate of mice treated as shown. GAPDH was used as a loading control. INS: insulin. Data are shown as mean ± standard error of the mean (SEM).

**Figure 7 cells-12-00443-f007:**
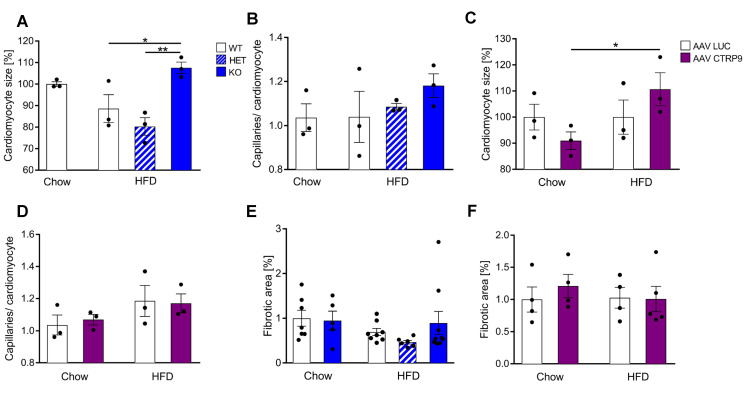
Histological analyses in mouse hearts with CTRP9 knock-out and overexpression**.** (**A**) Cardiomyocyte size measured as cross-sectional area and capillary density, measured as capillaries/cardiomyocyte ratio (**B**) in the indicated mice. WT: wild-type mice; KO homozygous and HET heterozygous CTRP9 knock-out mice. HFD: high-fat diet. (**C**) Cardiomyocyte size and capillary density (**D**) in the indicated mice treated with either AAV Luc (control) or AAV CTRP9 (leading to myocardial CTRP9 overexpression). (**E**,**F**) Quantification of cardiac fibrotic area in the indicated mice. Data are shown as mean ± standard error of the mean (SEM). * *p* < 0.05, ** *p* < 0.01.

**Figure 8 cells-12-00443-f008:**
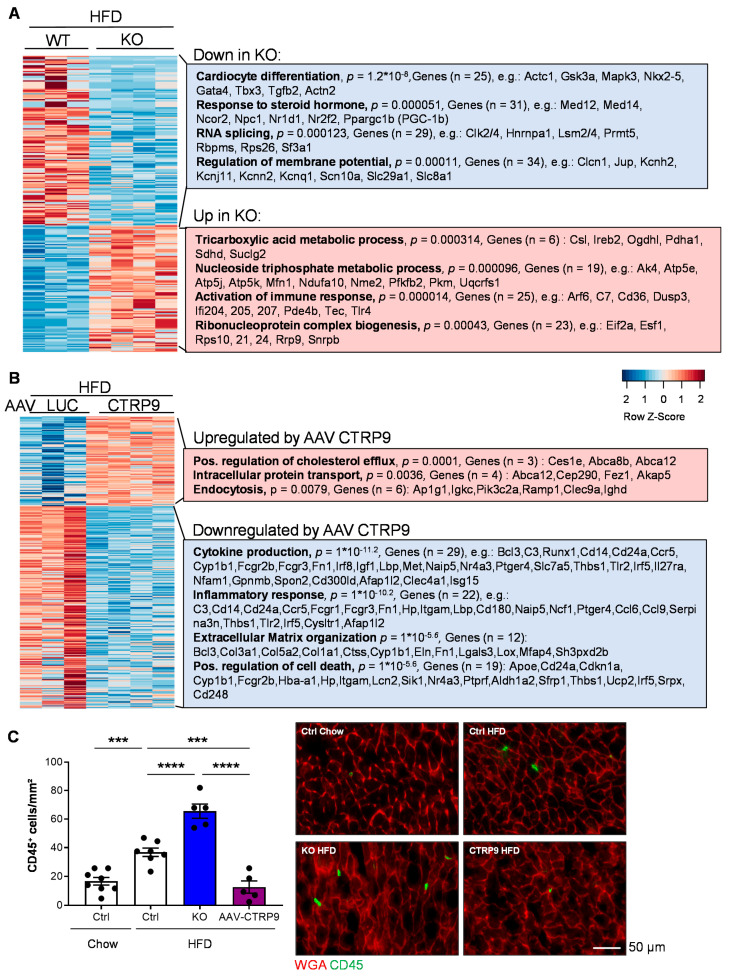
CTRP9 acts as an anti-inflammatory mediator in mouse hearts during high-fat diet. (**A**) Transcriptomic profiling from hearts of wild-type (WT) and CTRP9 knock-out (KO) mice after 12 weeks of high-fat diet (HFD). A heat map of differentially expressed genes is shown. Gene-ontology classes of differentially downregulated and upregulated genes and example genes are shown on the right. (**B**) Transcriptomic profiling from the hearts of AAV-Luc- and AAV-CTRP9-treated mice after 12 weeks of HFD feeding. Gene-ontology classes of differentially downregulated and up-regulated genes and example genes are shown on the right. (**C**) Quantification of the myocardial abundance of CD45+ leukocytes in the indicated mice and representative images. *** *p* < 0.001, **** *p* < 0.0001.

## Data Availability

The data presented in this study are available on request from the corresponding author. RNAseq data sets are deposited in the National Center for Biotechnology Information’s Gene Expression Omnibus database with accession number GSE219101.

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
