# Peer review of "C1q and Tumor Necrosis Factor Related Protein 9 Protects from Diabetic Cardiomyopathy by Alleviating Cardiac Insulin Resistance and Inflammation"

_cells, 2023, doi:10.3390/cells12030443_

Round 1
Reviewer 1 Report
The research article entitled, “C1q-TNF-Related Protein-9 protects from diabetic cardiomyopathy by alleviating cardiac insulin resistance and inflammation" by Haustein et al., aims to understand the impact of CTRP9 on early stages of diabetic cardiomyopathy induced by high fat diet using CTRP9 knock-out mice and AAV9 mediated cardiac CTRP9 overexpression. Authors reported that CTRP9 knock-out mice aggravated insulin resistance and triggered diastolic left ventricular dysfunction, whereas AAV9 mediated cardiac CTRP9 overexpression ameliorated cardiomyopathy. Moreover, extensive RNA sequencing from myocardial tissue of CTRP9 overexpressing and knock-out as well as respective control mice revealed that CTRP9 acts as anti-inflammatory mediator in the myocardium. In conclusion they reported that endothelial derived CTRP9 plays a prominent paracrine role to protect against diabetic cardiomyopathy.
Overall, this is a well written, significant and well-timed research article, this reviewer has certain recommendations that would assist to produce a more comprehensive overview of the topic:
Comments:
1, Immune cells also play a crucial role during diabetic Cardiomyopathy. Did authors find any significant difference in adaptive immune cells such as CD4 and CD8 etc? author should write a paragraph about immune cells and their effect on diabetic Cardiomyopathy with relation to their study (PMID: 16751419, PMID: 36093172, PMID: 36465455, PMID: 36337927 etc).
2, Authors can include the limitations to their study.
3, At least one illustrative figure may be provided as to highlight the summary of this study.
4, On page number 1 of this manuscript check the authors information. For example, there is an affiliation 6, but no author is mentioned form that affiliation. Affiliation 7 is incomplete.
5, instead of +++/***, exact p value can be place in graphs. And in all figure legends also account the statical test performed.
6, in figure 3, no mention of subfigure “F”.
7, The authors should cross-check all abbreviations in the manuscript. Initially, define in full name followed by abbreviation.
Author Response
We want to thank this reviewer for her/his valuable suggestions that helped us to improve our manuscript. The reviewer´s remarks and suggestions are shown in bold, while our response appears in normal fond.
Comments and Suggestions for Authors
The research article entitled, “C1q-TNF-Related Protein-9 protects from diabetic cardiomyopathy by alleviating cardiac insulin resistance and inflammation" by Haustein et al., aims to understand the impact of CTRP9 on early stages of diabetic cardiomyopathy induced by high fat diet using CTRP9 knock-out mice and AAV9 mediated cardiac CTRP9 overexpression. Authors reported that CTRP9 knock-out mice aggravated insulin resistance and triggered diastolic left ventricular dysfunction, whereas AAV9 mediated cardiac CTRP9 overexpression ameliorated cardiomyopathy. Moreover, extensive RNA sequencing from myocardial tissue of CTRP9 overexpressing and knock-out as well as respective control mice revealed that CTRP9 acts as anti-inflammatory mediator in the myocardium. In conclusion they reported that endothelial derived CTRP9 plays a prominent paracrine role to protect against diabetic cardiomyopathy.
Overall, this is a well written, significant and well-timed research article, this reviewer has certain recommendations that would assist to produce a more comprehensive overview of the topic:
We would like to thank this reviewer for her/his positive comments with regard to our study.
Comments:
1, Immune cells also play a crucial role during diabetic Cardiomyopathy. Did authors find any significant difference in adaptive immune cells such as CD4 and CD8 etc? author should write a paragraph about immune cells and their effect on diabetic Cardiomyopathy with relation to their study (PMID: 16751419, PMID: 36093172, PMID: 36465455, PMID: 36337927 etc).
Thanks for this comment. Unfortunately, we have not investigated the leukocyte subtype that is suppressed by CTRP9. As this reviewer suggests, we have now included a new paragraph about immune cells in diabetic cardiomyopathy in the revised manuscript (page 20, lines 579-587): “Although we have not further characterized the type of inflammatory cell that is sup-pressed by CTRP9, previous studies in diabetic cardiomyopathy have mainly implicat-ed macrophages and T-cells [39]. For instance, proinflammatory M1 macrophages were reported to be more abundant and contribute to cardiac damage in diabetic heart fail-ure [40-42]. Similarly, especially CD4 positive Th1 cells are upregulated in the periph-eral blood and hearts of diabetic patients and might contribute to cardiac damage [43-45]. In addition, neutrophils and B-cells could play a role, but more work is needed to define the role of specific immune cell subsets in diabetic cardiomyopathy [39].”
2, Authors can include the limitations to their study.
We included this in the revised version of our manuscript (page 20, lines 591-598):
“As limitations of our study, we a) did not decipher whether the protective effects of CTRP9 in diabetic cardiomyopathy depend primarily on its impact on cardiomyocyte metabolism or on its anti-inflammatory nature, b) we only investigated the effects of CTRP9 in one model of diabetic cardiomyopathy and c) only in male mice. Future studies should therefore investigate the effects of CTRP9 in additional disease models of DM (e.g. in db/db mice), also in female mice as well as address the primary benefi-cial target of CTRP9 more in depth.”
3, At least one illustrative figure may be provided as to highlightt the summary of this study.
We now uploaded a graphical abstract accompanying the revised version of our manuscript.
4, On page number 1 of this manuscript check the authors information. For example, there is an affiliation 6, but no author is mentioned form that affiliation. Affiliation 7 is incomplete.
We are sorry for these mistakes, which now have been corrected.
5, instead of +++/***, exact p value can be place in graphs. And in all figure legends also account the statical test performed.
Thanks for this comment. However, for the reason of clarity with multiple experimental groups, we prefer to indicate statistical significance by symbols. The statistical tests used are indicated on page 6, lines 254-257.
6, in figure 3, no mention of subfigure “F”.
We corrected this.
7, The authors should cross-check all abbreviations in the manuscript. Initially, define in full name followed by abbreviation.
We went over the whole manuscript again and improved the explanation of abbreviations at multiple sites in the text of the revised manuscript.
Reviewer 2 Report
This manuscript describes the use of high-fat diet to induce a pre-diabetic state in wt and ctrp9 ko mice, which results in weight gain as expected with an exaggerated level of glucose intolerance in HFD-treated KO mice with reduced diastolic function. The hearts of KO mice appeared to have reduced glucose uptake, despite higher circulating glucose levels and increased insulin levels. AAV-mediated overexpression of Ctrp9 also resulted in impaired glucose tolerance, but now with improved diastolic function. The authors performed some exploratory work to identify mechanistic insights into how Ctrp9 might contribute to diabetic cardiomyopathy (or just cardiac dysfunction in the setting of HFD).
Although the mechanistic insights provided by this manuscript are mainly limited to the findings on mitochondrial function, the overall message of the manuscript that Ctrp9 is an important regulator of (cardiac) metabolism in the setting of obesity is an important one. I only have some minor comments to improve the readability of the manuscript.
1. Are WT and KO mice used in Figure 3 exposed to HFD prior to PET-CT? Since only mice after HFD show differences in glucose tolerance, I would assume that they were HFD fed, but based on the images, the mice look lean.
2. Same question for the control and Ctrp9 AAV treated animals in figure 6.
3. Can the authors explain why both deletion and overexpression of Ctrp9 result in glucose intolerance in the setting of HFD?
4. How come there is an increased level of circulating insulin? Again, this appears to be the case for both KO and overexpression of Ctrp9. Some discussion of these findings would be important.
Author Response
We want to thank this reviewer for her/his valuable suggestions that helped us to improve our manuscript. The reviewer´s remarks and suggestions are shown in bold, while our response appears in normal fond.
Comments and Suggestions for Authors
This manuscript describes the use of high-fat diet to induce a pre-diabetic state in wt and ctrp9 ko mice, which results in weight gain as expected with an exaggerated level of glucose intolerance in HFD-treated KO mice with reduced diastolic function. The hearts of KO mice appeared to have reduced glucose uptake, despite higher circulating glucose levels and increased insulin levels. AAV-mediated overexpression of Ctrp9 also resulted in impaired glucose tolerance, but now with improved diastolic function. The authors performed some exploratory work to identify mechanistic insights into how Ctrp9 might contribute to diabetic cardiomyopathy (or just cardiac dysfunction in the setting of HFD).
Although the mechanistic insights provided by this manuscript are mainly limited to the findings on mitochondrial function, the overall message of the manuscript that Ctrp9 is an important regulator of (cardiac) metabolism in the setting of obesity is an important one. I only have some minor comments to improve the readability of the manuscript.
We want to thank this reviewer for her/his positive comments.
- Are WT and KO mice used in Figure 3 exposed to HFD prior to PET-CT? Since only mice after HFD show differences in glucose tolerance, I would assume that they were HFD fed, but based on the images, the mice look lean.
Yes, indeed both WT and KO mice received HFD before PET-CT. This is now more clearly indicated in the Figure legend. The size of the mice probably cannot be well judged from PET-CT images, especially as chow fed mice are not shown in comparison. However, the increase in body size is clearly evident from body weight measures (Figure 1B).
- Same question for the control and Ctrp9 AAV treated animals in figure 6.
Yes, also here AAV-control and AAV-CTRP9 treated mice both received HFD.
- Can the authors explain why both deletion and overexpression of Ctrp9 result in glucose intolerance in the setting of HFD?
Thanks for this comment. Although we cannot fully explain this seemingly paradoxical finding, we expanded the discussion on this point (page 18, line 519- page 19, line 539): “Unexpectedly, however, we observed that cardiac CTRP9 overexpression induced a state of systemic insulin resistance with increased serum insulin levels. This is in contrast to systemic transgenic CTRP9 overexpression in mice, which protected from diet induced obesity and metabolic dysfunction by triggering reduced food intake, an in-creased oxygen consumption rate in part due to enhanced mitochondrial content, in-creased fatty acid oxidation enzyme expression and chronic AMPK activation in skeletal muscle [9]. In these transgenic mice, CTRP9 was overexpressed in heart, skeletal muscle and the brain. Therefore, cardiac CTRP9 overexpression in our study might not be sufficient to alleviate systemic metabolic dysfunction, because especially skeletal muscles serve this purpose. In addition, overexpression in cardiomyocytes in our study is to a certain degree unphysiological, because CTRP9 is mainly derived from endothelial cells in the heart. However, due to the secreted nature of the protein, the originating cell might be less relevant. Why the metabolic state even worsens due to cardiac CTRP9 overexpression, currently remains unresolved, although counter regulatory effects of systemic endocrine systems due to altered cardiac metabolism could play a role. In this regard, it had been demonstrated that the heart can regulate metabolic activity and energy expenditure in peripheral tissues, for example by releasing endocrine (“cardiocrine”) factors, such as natriuretic peptides or MG53 [27-29]. Therefore, although we have not investigated this, an improved cardiac insulin responsiveness due to selective myocardial CTRP9 overexpression could modify cardiocrine signaling in a way that leads to insulin resistance in peripheral tissues and increased systemic insulin levels.”
- How come there is an increased level of circulating insulin? Again, this appears to be the case for both KO and overexpression of Ctrp9. Some discussion of these findings would be important.
Systemic insulin levels increase as a sign of systemic insulin resistance. For a deeper explanation of this point, please see our response to point 3).
Round 2
Reviewer 1 Report
The research article entitled "C1q-TNF-Related Protein-9 protects from diabetic cardiomyopathy by alleviating cardiac insulin resistance and inflammation" has improved to my satisfaction. The authors have incorporated all of the comments made.
Author Response
Thank you for your evaluating the paper and accepting it.
Reviewer 2 Report
No further comments
Author Response

(The authors gave the same response as above.)
